

# The correlation of region-specific lifestyle and subjective perception of oral health with oral health-related quality of life among Tibetan early adolescents in Ganzi: a cross-sectional study

Shaoying Duan*, Renjie Tang*, Chenchen Zhang, Qianqian Su, Huiyu Yang, He Cai and Tao Hu

State Key Laboratory of Oral Diseases & National Center for Stomatology & National Clinical Research Center for Oral Diseases & Frontier Innovation Center for Dental Medicine Plus, West China Hospital of Stomatology, Sichuan University, Chengdu, Sichuan, China
* These authors contributed equally to this work.

Corresponding authors
He Cai, caihe.dr@scu.edu.cn
Tao Hu, hutao@scu.edu.cn

## ABSTRACT

**Background:** The oral health-related quality of life (OHRQoL) of Tibetan adolescents has been largely overlooked.

**Aim:** This cross-sectional study examined the association of region-specific lifestyle, subjective perception, and clinician conditions of oral health with Tibetan adolescents' OHRQoL in Ganzi, Sichuan.

**Methods:** The OHRQoL was measured using standardized Child Oral Impacts on Daily Performances (sC-OIDP) scores. Binary logistic regression was used to explore the association between region-specific lifestyle, subjective perception of oral health, clinician oral health conditions (gingival bleeding on probing (BOP), dental calculus (DC), and decayed, missing, and filled teeth (DMFT)), and OHRQoL.

**Results:** In total, 485 Tibetan adolescents were included. In relation to region-specific lifestyle, the factors associated with poorer OHRQoL were residence altitude of 3,300 m (compared to 1,400 m), buttered tea consumption by adolescents or mother, and being a boarding student. Regarding the subjective perception of oral health, adolescents that rated bad or very bad oral health or self-reported dental pain had poorer OHRQoL. Interestingly, clinician oral health conditions were not related to OHRQoL.

**Conclusions:** This study underscores the relationship between region-specific lifestyle, subjective oral health perceptions, and OHRQoL among Tibetan adolescents. Addressing these factors through tailored health initiatives can play a pivotal role in improving oral health outcomes and overall quality of life in remote regions like Ganzi. Future research should focus on longitudinal studies to better understand the causality and long-term impact of targeted interventions.

## INTRODUCTION

Early adolescence is the crucial period for the development and maintenance of oral health. Oral health problems during adolescence not only negatively impact physical health but also psychological and social well-being (*Tuchtenhagen et al., 2021*). Oral health-related quality of life (OHRQoL) has become an increasingly important outcome for assessing individuals' comfort and satisfaction with their oral health (*Leao & Sheiham, 1995*). It is a multidimensional construct that encompasses the impacts of oral health on an individual's physical, psychological, and social well-being. A systematic review on OHRQoL in adolescents with 43 articles and a total of 28,267 11–18-year-olds found that the prevalence of oral impact was high in all the included studies (*Purohit et al., 2024*). Additionally, according to the 4[th] National Oral Health Survey (NOHS) of China, 76.6% of 89,582 adolescents aged 12–15 years reported oral impacts (*Wu et al., 2021a*). Oral impacts on daily performances were common among adolescents. While adolescents in China report relatively mild cases, it is important to note that OHRQoL can be influenced by differences in sample characteristics and the unique oral health challenges faced by specific populations (*Moghaddam et al., 2020*). Thus, special attention should be given to the severity of oral impaction in populations with distinct cultural and environmental habits.

According to the Seventh National Population Census of 2020, there are fewer than 8 million Tibetans in China. As a minority ethnic group, the Tibetan population accounts for a mere 0.5% of the whole population. Ganzi Tibetan Autonomous Prefecture is located at the southeastern edge of the Kang-Tibet Plateau to the west of Sichuan Province, China. With an average altitude of 3,500 m above sea level (*Wu et al., 2021c*), Ganzi is characterized by harsh natural conditions, cold climate, and low oxygen, resulting in distinct lifestyles. To accommodate the cold climate, local Tibetans prefer to have a high-fat and high-sodium drink, buttered tea, which is a traditional Tibetan food made from tea leaves, yak butter, water, and salt. In recent decades, large numbers of young adult couples have migrated from Ganzi to seek employment in large cities, leaving behind their children in the care of other family members (*Fellmeth et al., 2018*). Those left-behind children often board at school and experience separation from their parent(s) for more than 6 months.

The disparities of oral health present among adolescents of different ethnicities in certain areas may be a result of inequalities in medical and health service (*Zhang et al., 2019b*). In 2016, Ganzi Prefecture had only 1,279 practicing doctors, of which approximately 5% were dental professionals, amounting to just 63 dental practitioners serving a population of 1.1 million (*National Health and Family Planning Commission, 2018*; *Ganzi Statistical Yearbook, 2019*). In contrast, Beijing had the highest ratio of dental practitioners in China, with 3.83 per 10,000 people (*Liu, Xie & Shu, 2019*). This stark shortage of oral healthcare professionals severely limits Ganzi Prefecture's oral health development. The knowledge, attitudes, and behaviors of Tibetan adolescents regarding oral health are also concerning (*Hou et al., 2014*; *Yang, Lin & Lu, 2017*). According to the 4[th] NOHS of China, only 8.74% of Tibetan adolescents aged 12–15 brushed their teeth twice or more daily, and over 85% were unfamiliar with fluoridated toothpaste and dental

floss (*Wu et al., 2021b*). It was observed that Tibetan adolescents seemed to be susceptible to dental fluorosis and periodontal disease (*Wu et al., 2021b*), which might result from their region-specific lifestyle including living at high altitudes or drinking buttered tea (*Wu et al., 2021c*; *Zhang et al., 2019a*).

Poor oral health may weaken self-confidence, reduce self-evaluation, and deeply affect quality of life (*Kaur et al., 2017*; *Militi et al., 2021*). However, *Biazevic et al. (2008)* found no significant effect of periodontal disease on adolescents' quality of life in their study. As chronic periodontal disease often has no initial signs and symptoms, it may not affect OHRQoL (*Bernabé & Marcenes, 2010*). Furthermore, children's concerns about dental aesthetics and functionality are shaped by the social and cultural environments in which they live (*Mtaya, Astrom & Brudvik, 2008*). Despite these insights, existing research has largely overlooked the subjective perceptions of oral health and its relationship with OHRQoL in this ethnic minority group.

Region-specific lifestyle, subjective perception, and clinician conditions of oral health may be associated with OHRQoL among Tibetan adolescents, although evidence of these associations is limited. Hence, to improve our understanding of the OHRQoL in Tibetan adolescents, this article sought to assess the effects of region-specific lifestyle, subjective perception, and clinician conditions of oral health on OHRQoL of Tibetan adolescents in Ganzi.

# MATERIALS AND METHODS

## Study design

Portions of this text were previously published as part of a preprint (https://www.researchsquare.com/article/rs-3071403/v1). The current study followed the Strengthening the Reporting Observational Studies in Epidemiology (STROBE) guidelines. All the data used in the study were extracted from the cross-sectional survey of Tibetan adolescents conducted in Ganzi Tibetan Autonomous Prefecture, September 2016. Ethics approval of this survey was obtained from the Ethics Committee of the West China Hospital of Stomatology, Sichuan University (WCHSIRB-OT2016-077).

## Sampling and sample sizes

The sample size was determined based on the formula below:

$$n = \frac{\mu^2(1-p)}{\varepsilon^2 p(1 - nonresponse)}$$

in which $n$ is the sample size, $p$ (76.6%) is the impact prevalence among Chinese adolescents according to the 4[th] NOHS (*Wu et al., 2021b*), $\mu$(1.96) is the level of confidence, $\varepsilon$(0.1) is the margin of error, and the nonresponse rate is 20%. The formula above indicated that a minimum sample size of 147 adolescents was required.

A multistage stratified random-cluster sampling design was adopted in the survey to obtain representative participants of the Tibetan adolescents in Ganzi. Ganzi spans a wide range of altitudes, from 1,000 to 6,500 m above sea level (*Luo et al., 2019*). Since altitude affects physical health and lifestyle, and considering the harshness of the environment and
population density, three towns were randomly selected in the first stage to present three altitude ranges: 1,000–2,000 m, 2,000–3,000 m, and above 3,000 m (*Garvican-Lewis et al., 2015*; *Zhou et al., 2023*). This selection was made using the stratified sampling and probability-proportional-to-size (PPS) method (*Yin et al., 2017*). Those three towns were selected with an average altitude of 1,400, 2,560 and 3,300 m, respectively. In the second stage, two schools were randomly selected from each town using the PPS method. In the third stage, specific participants were chosen using quota sampling. Only students studying in junior middle school (expected to contain Tibetan adolescents aged 11–15 years) who could complete the questionnaire independently in the Chinese language were included in the survey. Written consent to conduct and publish the study was also obtained from adolescents' parent or statutory guardian. Adolescents who were only familiar with the Tibetan language or had serious systemic diseases were excluded. A total of 490 adolescents were included in the survey, four of which were excluded from the current study for unrecorded age and one for missing information of gender. Ultimately, 485 Tibetan adolescents were included in our study.

## Questionnaire and clinical examinations

All the participants were asked to fill a paper-based questionnaire individually in their classroom, which included closed-ended questions of demographic characteristics, socio-economic status, region-specific lifestyle, and subjective perception on oral health (Table S1), as well as the Chinese version of the Child Oral Impacts on Daily Performances (C-OIDP) scale which had been widely used among Tibetan people in the 4th NOHS of China.

Using the Community Periodontal Index (CPI) probes, mouth mirrors, mobile dental chairs, and portable lights, the examination of dental caries and gingival status was conducted in schools by three training examiners and one calibrating examiner according to the basic method of oral health survey recommended by the *World Health Organization (2013)*. Cohen's kappa coefficient was tested to measure the inter- or intra-examiner consistency. In the study, examiners' kappa coefficients were all >0.80 for dental caries and gingival health status. All the examiners were blinded to the results of questionnaires.

## Outcomes

OHRQoL was evaluated using the overall score of C-OIDP, the most commonly-used tool for assessing this measure in adolescents (*Haag et al., 2017*). This tool was comprised of eight items assessing the impact of oral health across three domains, physical (eating, speaking, cleaning teeth), psychological (sleeping, grinning, emotion), and social (studying, social contact), in the past six months with the following response options: "no impact" (zero point), "little impact" (one point), "moderate impact" (two points), and "severe impact" (three points). A transferred C-OIDP score (tC-OIDP) was calculated by adding the scores of the eight items, then divided by the maximum score (24 points) and multiplied by 100. The tC-OIDP score was then standardized using the mean and standard deviation (SD) of the tC-OIDP of adolescents in the whole Sichuan Provence (*Cheng et al., 2023*; *Jenkinson, Layte & Lawrence, 1997*):

$$\text{standardized score} = \frac{\text{transferred score} - \mu}{\sigma} * 10 + 50$$

in which $\mu$ (17.53) is the mean tC-OIDP in Sichuan Province and $\sigma$ (16.59) is the SD of tC-OIDP according to the oral health survey of Sichuan Province, 2015–2016. The standardized score facilitates intuitive interpretation of the results (compared with a reference value of 50) and enhances the extrapolation of the findings (*Fawcett & Neukrug, 2014*).

Higher standardized C-OIDP (sC-OIDP) scores indicated poorer OHRQoL, while lower sC-OIDP corresponded to better OHRQoL. Finally, the sC-OIDP was dichotomized, yielding the categories 'OHRQoL was better than the average of Sichuan (sC-OIDP < 50)' and 'OHRQoL was poor than the average of Sichuan (sC-OIDP > 50)'. This formula was also used to calculate the standardized score across three aspects and eight items. Cronbach's alpha was used for internal consistency reliability. In this study, the Cronbach's alpha was 0.75 (95% CI [0.71–0.78]).

### Exposure variables

Clinician oral health variables, including decayed, missing, and filled teeth (DMFT) index, gingival bleeding on probing (BOP), and dental calculus (DC) were assessed by examiners *via* CPI probes. DMFT index represents the sum of the number of teeth decayed (*i.e.*, with an unmistakable cavity, undermined enamel, detectably softened floor, or wall of a tooth with/without a filling or seal), missing due to caries, and filled in the permanent dentition including the third molar. BOP and DC were studied as a dichotomous variable (*da Fonseca et al., 2020*): "Absence/Presence of gingival bleeding or calculus deposit".

Region-specific lifestyle variables, including residence altitude ("1,400 m/2,560 m/3,300 m"), buttered tea consumption by adolescents themselves in childhood ("Don't drink/Drink") and their mothers ("Don't drink/Drink/Unknown or orphan"), and whether the adolescent was a boarding student in the most recent year ("No/Yes"). Relevant data of residence altitude were recorded by the data entry staff based on the participants' permanent residence addresses filled in the questionnaire.

Subjective perception of oral health variables included self-rated oral health ("Good or very good/Fair/Poor or very poor") and self-reported dental pain in the last 12 months ("No/Yes/Unknown").

Based on the conceptual theoretical model introduced to epidemiologic research by *Greenland, Pearl & Robins (1999)*, we constructed a directed acyclic graph (DAG) to illustrate the hypothesized impact of exposure factors on OHRQoL (Fig. 1). This model included participants' socioeconomic characteristics as confounders that may influence both exposure factors and OHRQoL.

### Confounders

The following demographic variables were taken as covariates in the study: age, gender ("Male/Female"), household type ("Non-agricultural family/Agricultural family"), and

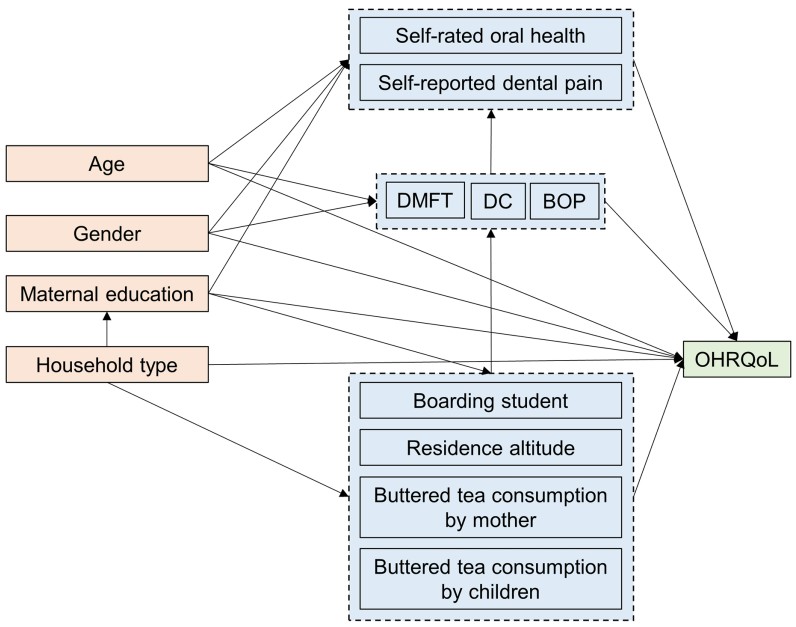

**Figure 1  Directed acyclic graphic depicting the association between exposure and oral health-related quality of life.**               

maternal education ("High" (Tertiary or bachelor's degree or above), "Medium" (High school), "Low" (Junior high school or below), "Other" (Unknown or orphan)).

## Statistics

Descriptive analysis was used to characterize the sample and distribution of each category of Tibetan adolescents in Ganzi, Sichuan. Continuous variables were described using mean and SD, and categorical variables using absolute and relative values. Bivariate analysis was performed using parametric tests (2-sample t-test or one-way ANOVA) to assess the association of those variables, residence altitude, and household type. Binary logistic regression analysis was conducted to determine whether region-specific lifestyle, subjective perception, and clinician oral health conditions were associated with oral impacts among the Tibetan adolescents, and was adjusted for age, gender, household type, and maternal education. The binary logistic regression model relies on assumptions including independent observations and absence of multicollinearity (*Harris, 2021*). Prior to analyses, the assumptions were verified. Crude and adjusted odds ratio (OR) values and respective 95% CIs were calculated as measures of association. Data were analyzed by SPSS version 20.0 (IBM Corporation, Armonk, NY, USA) and were presented as SD. *P* values less than 0.05 were considered statistically significant (2-tailed test). All available data were used in the analyses.

## RESULTS

### Descriptive statistics

The response rate was 99% (485/490) with a mean age of 12.11 ± 0.54. Of the 485 Tibetan adolescents, 225 (46.4%) were boys. The proportion of agricultural families was higher

(83.3%) than non-agricultural families (16.7%). Most mothers (60.8%) only had a junior high school degree or less. The majority of Tibetan adolescents (90.9%) and their mothers (90.9%) drank buttered tea. In terms of oral health conditions, more than half had dental pain (64.5%), BOP (75.5%), and DC (82.7%).

The mean tC-OIDP of the adolescents was 26.50 ± 18.57. After standardization, the mean sC-OIDP of the participants was 55.41 ± 11.19 with a median score of 54.50 and range from 39.44 to 99.70. Among those participants, 288 (59.4%) adolescents had a sC-OIDP higher than 50 which indicated a relatively poor OHRQoL. The distribution of adolescent demographic characteristics, region-specific lifestyle, subjective perception of oral health, and sC-OIDP is shown in Table 1.

The frequency distribution of three domains and eight items of oral impact performance among Tibetan adolescents was shown in Fig. 2. More than half had oral impacts on daily performances in physiological, psychological, and social domains. As for the eight items, emotion was the most commonly affected (sC-OIDP > 50, 62.7%), followed by cleaning teeth (sC-OIDP > 50, 56.1%), and grinning (sC-OIDP > 50, 54.8%), while oral impact on eating was the slightest impacted item (sC-OIDP > 50, 36.7%).

## Logistic regression

In the binary logistic regression analysis, adjusted by age, gender, household type, and maternal education, a total of seven variables including residence altitude of 3,300 m (OR = 1.880, 95% CI [1.101–3.209], $P = 0.021$), buttered tea consumption by adolescents (OR = 2.284, 1.200–4.348, $P = 0.012$) or mother (OR = 2.278, 1.171–4.432, $P = 0.015$), being a boarding student (OR = 1.761, 1.180–2.626, $P = 0.006$), poor or very poor of self-rated oral health (OR = 4.360, 2.341–8.120, $P < 0.001$), and self-reported dental pain (OR = 2.836, 1.836–4.382, $P < 0.001$) showed a significant negative association with sC-OIDP. Interestingly, clinician conditions, such as BOP, DC, and DMFT were not related to OHRQoL (Table 2).

The associations of exposure variables, three domains, and all eight items assessing the impact of oral health are displayed in Tables S2 and S3. Clinician oral health was not correlated with any items of oral impact or any domains except that DMFT was correlated with oral impact in the social domain (OR = 0.88, 95% CI [0.78–1.00], $P < 0.05$). Additionally, buttered tea consumption by children or mother was not associated with any items or domains of oral impact ($P > 0.05$). Oral impact on eating was the most significant item after comparing tC-OIDP (40.76 ± 32.13). However, after standardization, studying was the most seriously affected performance with a mean sC-OIDP of 54.72 ± 14.11, followed by emotion (53.78 ± 11.64) and sleeping (53.45 ± 12.89) (sC-OIDP > 50, 54.8%), while oral impact on eating became the last seriously impacted item (52.50 ± 11.08).

## DISCUSSION

This is the first study to assess the OHRQoL among Tibetan adolescents in Ganzi, Sichuan. Overall, the study revealed a relatively poorer OHRQoL of Tibetan adolescents in Ganzi compared to the average of adolescents in Sichuan. Also, it was observed that region-specific lifestyles such as living at a high altitude, mother or adolescents drinking buttered

**Table 1 Demographic characteristics of the Tibetan adolescents in Ganzi, Sichuan.**

| Variables | | All participants | tC-OIDP | Participants with sC-OIDP < 50 | Participants with sC-OIDP > 50 |
|---|---|---|---|---|---|
| | | Mean ± SD/count (%) | Mean ± SD | Mean ± SD/count (%) | Mean ± SD/count (%) |
| Age | | 12.11 ± 0.54 | 26.50 ± 18.57 | 12.12 ± 0.59 | 12.10 ± 0.50 |
| Gender | Male | 225 (46.4) | 28.38 ± 19.22 | 84 (37.3) | 141 (62.7) |
| | Female | 260 (53.6) | 24.87 ± 17.86 | 113 (43.5) | 147 (56.5) |
| Household type | Non-agricultural family | 81 (16.7) | 27.95 ± 19.42 | 31 (38.3) | 50 (61.7) |
| | Agricultural family | 404 (83.3) | 26.21 ± 18.40 | 166 (41.1) | 238 (58.9) |
| Maternal education | Low | 295 (60.8) | 27.28 ± 18.85 | 113 (38.3) | 182 (61.7) |
| | Medium | 32 (6.6) | 24.24 ± 19.51 | 15 (46.9) | 17 (53.1) |
| | High | 55 (11.3) | 19.81 ± 15.21 | 33 (60.0) | 22 (40.0) |
| | Other | 103 (21.2) | 28.54 ± 18.49 | 36 (35.0) | 67 (65.0) |
| Residence altitude | 1,400 m | 180 (37.1) | 25.72 ± 17.97 | 74 (41.1) | 106 (58.9) |
| | 2,560 m | 194 (40.0) | 23.33 ± 16.57 | 93 (47.9) | 101 (52.1) |
| | 3,300 m | 111 (22.9) | 33.32 ± 21.07 | 30 (27.0) | 81 (73.0) |
| Buttered tea consumption by children | Don't drink | 44 (9.1) | 22.24 ± 18.53 | 26 (59.1) | 18 (40.9) |
| | Drink | 441 (90.9) | 26.93 ± 18.54 | 171 (38.8) | 270 (61.2) |
| Buttered tea consumption by mother | Don't drink | 41 (8.4) | 23.56 ± 17.54 | 24 (58.5) | 17 (41.5) |
| | Drink | 441 (90.9) | 26.76 ± 18.60 | 172 (39.0) | 269 (61.0) |
| | Unknown or orphan | 3 (0.6) | 27.99 ± 30.68 | 1 (33.3) | 2 (66.7) |
| Boarding student | No | 278 (57.3) | 23.20 ± 16.97 | 132 (47.5) | 146 (52.5) |
| | Yes | 207 (42.7) | 30.93 ± 19.70 | 65 (31.4) | 142 (68.6) |
| Self-rated oral health | Good or very good | 116 (23.9) | 20.74 ± 17.28 | 63 (54.3) | 53 (45.7) |
| | Fair | 272 (56.1) | 25.82 ± 17.97 | 114 (41.9) | 158 (58.1) |
| | Poor or very poor | 97 (20.0) | 35.30 ± 18.67 | 20 (20.6) | 77 (79.4) |
| Self-reported dental pain | No | 125 (25.8) | 20.02 ± 17.54 | 71 (56.8) | 54 (43.2) |
| | Yes | 313 (64.5) | 29.51 ± 18.49 | 103 (32.9) | 210 (67.1) |
| | Unknown | 47 (9.7) | 23.71 ± 17.18 | 23 (48.9) | 24 (51.1) |
| BOP | No | 119 (24.5) | 26.57 ± 19.72 | 49 (41.2) | 70 (58.8) |
| | Yes | 366 (75.5) | 26.48 ± 18.20 | 148 (40.4) | 218 (59.6) |
| DC | No | 84 (17.3) | 28.26 ± 20.86 | 32 (38.1) | 52 (61.9) |
| | Yes | 401 (82.7) | 26.13 ± 18.06 | 165 (41.1) | 236 (58.9) |
| DMFT | | 0.87 ± 1.49 | 26.50 ± 18.57 | 1.03 ± 1.42 | 0.77 ± 1.53 |
| sC-OIDP | | 55.41 ± 11.19 | | 44.98 ± 3.19 | 62.54 ± 8.85 |

**Note:**
tC-OIDP, transferred Child Oral Impacts on Daily Performances score; sC-OIDP, the Child Oral Impacts on Daily Performances score that standardized to the Sichuan adolescent mean score; SD, standard deviation; BOP, gingival bleeding on probing; DC, dental calculus; DMFT, decayed, missing, and filled teeth.

tea, boarding at school, and subjective perception of oral health including self-rated poor oral health and self-reported dental pain were associated with a poorer OHRQoL. However, there was no significant association between clinician oral conditions, such as BOP, DC, DMFT, and OHRQoL.

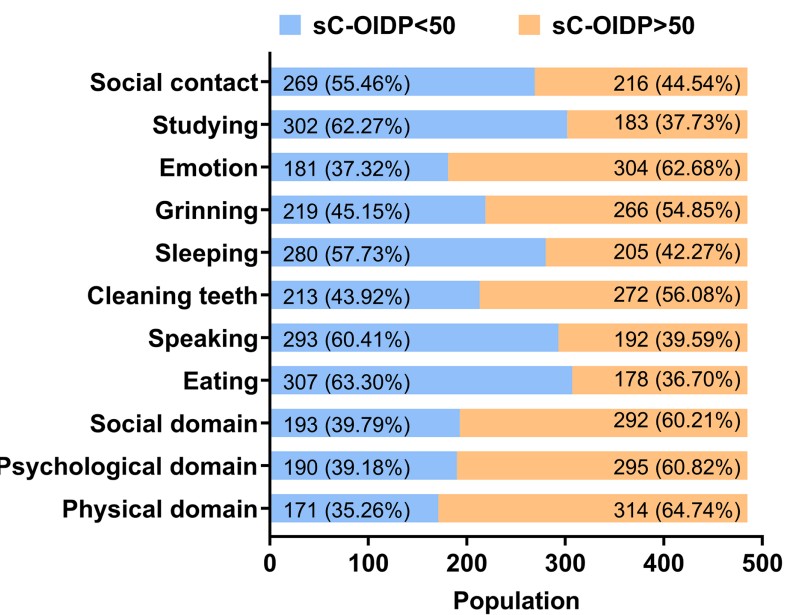

**Figure 2** Frequency distribution of three domains and eight items of oral impact performance.

The main scales for assessing children's OHRQoL include C-OIDP, the Child Perceptions Questionnaire (CPQ), and the Child Oral Health Impact Profile (COHIP) (*Gilchrist et al., 2014*; *Laborne et al., 2024*). The Chinese version of the CPQ is divided into CPQ8-10 and CPQ11-14 based on age (*Kong, Gao & Yuan, 2023*; *Yau et al., 2015*). Given the wide age range of Tibetan adolescent junior high school students (11–15 years old), the CPQ scale was not suitable for this study. Additionally, both the CPQ and COHIP scales contain many questions, making them less practical for large-scale epidemiological surveys (*Alvarez-Azaustre, Greco & Llena, 2024*). Given the verified validity and reliability of the C-OIDP among Chinese adolescents in China and its relatively few items, this study chose this scale to assess OHRQoL (*Gilchrist et al., 2014*; *Hongxing et al., 2014*; *Liu et al., 2021b*; *Zhang et al., 2021*). Our study found the mean tC-OIDP of the adolescents was 26.50 (SD 18.57) which was significantly higher than found in the similar study in Guangxi Province of China (17.84) (*Zhang et al., 2021*), Turkey (13.11) (*Peker et al., 2020*), and Malaysia (5.45) (*Berhan Nordin et al., 2019*), suggesting that the OHRQoL of Tibetan adolescents in Ganzi was relatively poor. Differences in ethnicity, customs, culture, and geographical circumstance might explain the differences in C-OIDP scores between adolescents in different locations. After the transferred treatment, the highest impact was on eating in the current population, which corresponded to a multitude of previous studies (*Berhan Nordin et al., 2019*; *Zahra, Marhazlinda & Yusof, 2024*). Dental pain caused by decay and bleeding gingiva might interfere with biting ability and the taste of food (*Renton, 2011*). However, after the standardized treatment, the highest impact was on studying, while the last serious impact was on eating. This result suggests that, compared with adolescents in Sichuan Province, studying was the most seriously impacted performances in Ganzi. C-OIDP score standardization can highlight issues of particular locality and assist in targeted planning

**Table 2 Binary logistic regression analysis of the associations between sC-OIDP scores and predictors.**

| Variables | | OR$^{crude}$ (95% CI) | OR$^{adjusted\ \#}$ (95% CI) |
|---|---|---|---|
| Region-specific lifestyle | Residence altitude | | |
| | 1,400 m | Reference group | |
| | 2,560 m | 0.758 [0.504–1.142] | 0.900 [0.583–1.390] |
| | 3,300 m | 1.885 [1.128–3.150] | **1.880 [1.101–3.209]**\* |
| | Buttered tea consumption by children | | |
| | Don't drink | Reference group | |
| | Drink | 2.281 [1.214–4.286] | 2.284 [1.200–4.348]\* |
| | Buttered tea consumption by mother | | |
| | Don't drink | Reference group | |
| | Drink | 2.208 [1.153–4.230] | **2.278 [1.171–4.432]**\* |
| | Boarding student | | |
| | No | Reference group | |
| | Yes | 1.975 [1.356–2.877] | **1.761 [1.180–2.626]**\*\* |
| Subjective perception of oral health | Self-rated oral health | | |
| | Good or very good | Reference group | |
| | Fair | 1.647 [1.064–2.552] | 1.556 [0.997–2.429] |
| | Poor or very poor | 4.576 [2.480–8.445] | **4.360 [2.341–8.120]**\*\*\* |
| | Self-reported dental pain | | |
| | No | Reference group | |
| | Yes | 2.681 [1.752–4.101] | **2.836 [1.836–4.382]**\*\*\* |
| Clinician conditions | BOP | | |
| | No | Reference group | |
| | Yes | 1.031 [0.677–1.570] | 0.950 [0.617–1.464] |
| | DC | | |
| | No | Reference group | |
| | Yes | 0.880 [0.543–1.427] | 0.828 [0.505–1.356] |
| | DMFT | 0.889 [0.786–1.005] | 0.904 [0.798–1.025] |

Notes:
sC-OIDP, the Child Oral Impacts on Daily Performances score that standardized to the Sichuan adolescent mean score; OR$^{crude}$, crude odds ratio; OR$^{adjusted}$, adjusted odds ratio; CI, confidence interval; BOP, gingival bleeding on probing; DC, dental calculus; DMFT, decayed, missing, and filled teeth. # Adjusting for age, gender, household type and maternal education.
\* $P < 0.05$.
\*\* $P < 0.01$.
\*\*\* $P < 0.001$. All statistically significant results were highlighted in bold.

for interventions in certain areas. Notably, if the C-OIDP scores reported in different studies can be standardized based on regional or national data, the comparability across research will be greatly improved.

In this study, the regression analysis showed that region-specific lifestyle and subjective perception of oral health were the key predictors affecting the OHRQoL of Tibetan adolescents after controlling the confounders. Drinking buttered tea is a traditional and important dietary characteristic among Tibetan. The fluoride content in the drinking water of the Tibetan population in Ganzi Prefecture maintains levels below 1 mg/L (*Cai & Zhang, 2008*). Meanwhile, the water-dissolved fluoride concentrations in buttered tea

range from 533.89 to 617.32 mg/L, significantly higher than the fluoride concentration found in local drinking water (*Jin et al., 2000*). High fluoride concentration in buttered tea was the major reason for endemic dental fluorosis (*Do, Levy & Spencer, 2012*). As described in a previous study (*Zhang et al., 2019a*), mothers drinking buttered tea was a strong risk factor for children's fluorosis. It was observed that buttered tea consumption by mother or children was associated with OHRQoL. This finding might be explained by the fact that dental fluorosis causes changes in tooth color, which is an aesthetic concern of adolescents (*Do, Levy & Spencer, 2012*).

Living in high altitude areas was related to poor OHRQoL. The atmosphere at high altitudes is typically dry, cold, and hypoxic. In high altitude areas, residents often experience dry mouth and tongue (*Liu et al., 2021a*), which can be uncomfortable during daily life. Moreover, it has been reported that altitude is a risk factor to oral microbiota disorders (*Dong et al., 2021*). An imbalance in oral microbiota is associated with various diseases including dental caries, periodontitis, and halitosis (*Chew et al., 2024*; *Li et al., 2023*; *Zhu et al., 2023*). These may explain why living in a high-altitude environment is associated with worse OHRQoL.

When referring to other factors in region-specific lifestyle, boarding students were worthy of attention. Similarly, our results found that boarding students had poor life quality when compared with non-boarding students (*Küçük & Günay, 2014*). In China, parental migration has been a common phenomenon in remote areas within the last decade, causing adolescents to board at school without parental involvement. Studies have found correlation between parental involvement and OHRQoL (*Knorst et al., 2019*). Boarding schools can cause social and emotional distress and negatively impact mental health (*Kleinfeld & Bloom, 1977*; *Zhong, Feng & Xu, 2024*), which may amplify the impact of adolescent oral problems on their daily lives. Additionally, in the absence of parental guidance, adolescents may experience poorer oral health habits and have poor oral (*Qiu et al., 2018*) and psychological health (*Jia et al., 2010*).

Subjective perception of oral health was confirmed to affect OHRQoL; that is, poor self-rated oral health resulted in poor OHRQoL and adolescents who had the experience of dental pain also had a poor OHRQoL. The association was consistent with previous studies that reported similar observations (*Barasuol et al., 2020*; *Yin et al., 2023*; *Zahra, Marhazlinda & Yusof, 2024*) and might be explained by the following explanation: the concept of OHRQoL was proposed to measure people's comfort and satisfaction on their oral health, which is a subjective as well as a sophisticated index. Poor self-rated oral health may contribute to increased depressive symptoms (*de Souza Barbosa et al., 2016*). For clinicians, the link between dental pain and poorer OHRQoL is unsurprising (*Zhong, Feng & Xu, 2024*). Adolescents experiencing pain often face restricted chewing ability, sleep disturbances, and difficulties in daily activities and learning, which can negatively impact their social and emotional well-being.

Interestingly, clinician oral health variables including DMFT, BOP, and DC were not associated with adolescent OHRQoL. Although it was revealed that DMFT was associated with social domain, OR values ranged from 0.78–1.00 (1.00 inclusive) and $P$ values close to 0.05. Thus, it can be assumed that there is no statistical significance of the relationship

between DMFT and social domain. A study of Spanish adolescents yielded results similar to those found in our research (*Alvarez-Azaustre, Greco & Llena, 2024*). Other studies found that people with a higher prevalence of caries (DMFT > 0) were more likely to report C-OIDP score > 0 (*Kanungo et al., 2023*; *Simangwa et al., 2020*; *Zahra, Marhazlinda & Yusof, 2024*). The severity of dental caries was associated with dental pain, which was more likely to be associated with C-OIDP (*Anthony et al., 2023*; *Fernandez et al., 2024*). This study did not evaluate the severity of dental caries among Ganzi Tibetan adolescents. It is possible that the dental caries in this group were relatively mild and had little impact on their daily lives. A negative association between BOP, DC, and children's OHRQoL has been found in some studies (*Zhang et al., 2021*). However, previous studies showed that BOP or DC were not related to children's OHRQoL (*Liu et al., 2021b*), and a similar conclusion was reached in this study. One possible explanation for these counterintuitive results was that the perception of OHRQoL has a subjective component and thus, varies from one cultural background to another (*Chen & Hunter, 1996*). Direct comparisons with the published articles between different groups must be interpreted with caution. In addition, periodontal health was assessed by the presence or absence of BOP or DC. Collectively, the association between self-perceived and clinician oral health condition deserves further exploration based on more details of severity and extent of oral problems.

The finding of the present study should be interpreted within some limitations. The data used in the study were extracted from a cross-sectional survey in 2016, which hampered the inference of causality and reflected a single point in time. However, given that health service provision in Ganzi has remained relatively unchanged and oral health has received less attention in this region, the results of this study remain relevant and informative (*Statistics SpBo, 2024*; *Zhou, 2024*). Also, questionnaires were self-reported by Tibetan adolescents, which might have introduced response bias into the study in contrast to interview. However, since the OHRQoL we used included a psychological component, and the survey was conducted using the C-OIDP questionnaire with interpretation by a professional dentist, the self-reported data have been shown to align with interview results (*Rosel et al., 2010*; *Tsakos et al., 2008*). Moreover, given the high altitude and limited logistical resources in the Ganzi Tibetan region, self-reported data collection was particularly suitable for this study. Additionally, this survey conducted among school adolescents might have caused selection bias. The nomadic lifestyle of this ethnic group makes it challenging to track out-of-school adolescents. However, since local compulsory education covers children aged 6–16, the selection bias was likely minimal. Furthermore, given the study's focus on distinctive Tibetan dietary habits and parental involvement, the inclusion of school-based adolescents still provides valuable insights.

## CONCLUSIONS

In the present study, it was found that Tibetan adolescents living in Ganzi seemed to have a relatively poor OHRQoL. Region-specific lifestyle including residing at high altitude (3,000 m), their or their mothers' consumption of buttered tea, boarding at school, and subjective perception of oral health including poor oral health and dental pain were the key predictors affecting their OHRQoL. Local government should develop measures to reduce

inequalities in housing between low- and high-altitude regions and provide an adequate supply of clean drinking water. Easier access to dental care and regular check-ups are also needed to help local adolescents maintain good oral health and prevent dental pain. Meanwhile, communication and support from their parents and adolescent psychological assistance from schools and society may boost the self-perception of adolescents. Contrary to the findings of previous research, no relationship between DMFT and OHRQoL was detected in this study. Additional investigation is necessary for this issue.

## ACKNOWLEDGEMENTS

Special thanks to the Tibetan adolescents who participated in the oral health survey in Ganzi, and to the staff and teachers in local schools who assisted with recruitment.

### Funding

This work was supported by the Research and Development Program, West China Hospital of Stomatology Sichuan University (Grant number [No. LCYJ-ZD-202301]) and the National Natural Science Foundation of China (Grant number [No. 72104162]). The funders had no role in study design, data collection and analysis, decision to publish, or preparation of the manuscript.

### Grant Disclosures

The following grant information was disclosed by the authors:
West China Hospital of Stomatology Sichuan University: LCYJ-ZD-202301.
National Natural Science Foundation of China: 72104162.

### Competing Interests

The authors declare that they have no competing interests.

### Author Contributions

- Shaoying Duan performed the experiments, prepared figures and/or tables, authored or reviewed drafts of the article, and approved the final draft.
- Renjie Tang performed the experiments, prepared figures and/or tables, authored or reviewed drafts of the article, and approved the final draft.
- Chenchen Zhang performed the experiments, prepared figures and/or tables, and approved the final draft.
- Qianqian Su analyzed the data, prepared figures and/or tables, and approved the final draft.
- Huiyu Yang analyzed the data, prepared figures and/or tables, and approved the final draft.
- He Cai conceived and designed the experiments, authored or reviewed drafts of the article, and approved the final draft.
- Tao Hu conceived and designed the experiments, authored or reviewed drafts of the article, and approved the final draft.

## Human Ethics

The following information was supplied relating to ethical approvals (*i.e.*, approving body and any reference numbers):

Before the start of the study, the Ethics Committee of the West China Hospital of Stomatology, Sichuan University (WCHSIRB-OT2016-077), provided ethics approval for this study.

## Data Availability

The raw measurements are available in the Supplemental File.

## Supplemental Information

Supplemental information for this article can be found online at http://dx.doi.org/10.7717/peerj.18842#supplemental-information.

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
