# Peer review of "The correlation of region-specific lifestyle and subjective perception of oral health with oral health-related quality of life among Tibetan early adolescents in Ganzi: a cross-sectional study"

_PeerJ, doi:10.7717/peerj.18842_

## Round 0.1 · original submission · Major Revisions

The authors are requested to carefully revise the manuscript and answer the questions raised by the reviewers.

Reviewer 1 ·

Basic reporting

No comment

Experimental design

No comment

Validity of the findings

No comment

Additional comments

The manuscript is very interesting, well written. The introduction, methods, and results are well structured. The discussion is very informative, it addresses and justifies all the outcomes of the study.
I would like to thank the authors for this article.

Reviewer 2 ·

Basic reporting

How butter tea is going to influence the dental condition? Fluorosis could be due to high fluorine content in the water from which tea is made.

Experimental design

No comment

Validity of the findings

no comments

Reviewer 3 ·

Basic reporting

Clear and unambiguous, professional English used throughout.
Answer: Need scientific editing.
Literature references, sufficient field background/context provided.
Answer: Mostly local oriented informations and investigations are done.
Professional article structure, figures, tables. Raw data shared.
Answer: Scientific merits are lacking. Self reported data are used.
Self-contained with relevant results to hypotheses.
Answer: Not specifically.

Experimental design

The data used in the study were extracted from a cross-sectional survey in 2016, which hampered the inference of causality and reflected a single point in time.

Then, questionnaires were self-reported by Tibetan adolescents under explanation from qualified dentists in class, which might be introduced response bias into the study in contrast to interview.

Also, this survey conducted among school adolescents might cause selection bias because those out-of-school adolescents were excluded from the survey, despite their rather small percentage in the context of compulsory education. Even so, the investigation combined with Tibetan characteristic dietary habits and parents' involvement is an innovation deserves further exploring.

Validity of the findings

The data used in the study were extracted from a cross-sectional survey in 2016, which hampered the inference of causality and reflected a single point in time.

Then, questionnaires were self-reported by Tibetan adolescents under explanation from qualified dentists in class, which might be introduced response bias into the study in contrast to interview.

Also, this survey conducted among school adolescents might cause selection bias because those out-of-school adolescents were excluded from the survey, despite their rather small percentage in the context of compulsory education. Even so, the investigation combined with Tibetan characteristic dietary habits and parents' involvement is an innovation deserves further exploring.

Additional comments

1. Local investigation.
2. Poor scientific merit.
3. Lots of limitations.
4. Sample size not followed based on calculations.

Reviewer 4 ·

Basic reporting

English language needs to be corrected in line 2, 7, 140. It is advised for a native English speaker to proofread the manuscript.

Sentence from line4-8 is quite extensive and difficult to understand. It is better to divide it into 2 sentences for clarity.
Line 8-10: Which region was included in this review? Since your research is region-specific, it is important to mention region-specific studies in your literature review.
Both references mentioned in the introduction have contrasting results. While citing these, it is important to mention the reason behind contrast, and how is it relevant to your research.
Kindly rewrite the introductory paragraphs for more clarity.

Line 24: If the authors wish to provide context of Tibetan, they should give details about the oral hygiene awareness programs and oral health education in schools. The audience needs to know the general oral health status of Tibetan, please provide available literature in this regard.
Line 30-31: The gap is not sufficient. The rationale of the study. Is not clear. When the authors have already previously declared that Tibetan people are susceptible to oral diseases, do they expect different results? Explain
Line 32-33: Instead of a question statement, rephrase it to make a declarative statement.
Please add latest relevant literature to discuss in the discussion section. It is advisable to suggest reasons for the results.

Experimental design

Which adolescent group was targeted in this study? Early, middle or late adolescents?
Line 72: The authors need to explain who translated the questionnaire and who validated the questionnaire before use?
Line 85-86: Rephrase the sentence, avoid redundant use of OHRQoL
Line 100: The term sC-OIDP has not been explained previously, kindly explain it before using acronym
Line 114: Please close the bracket

Validity of the findings

no comment

Additional comments

It is a very interesting study with pertinent findings. I suggest the authors make necessary corrections and resubmit the draft with improved write-up.

Reviewer 5 ·

Basic reporting

No comment

Experimental design

No comments

Validity of the findings

No comments

Additional comments

Systematically well-written articles with a strong scientific background, detailed methodology, explanation of study design, sampling methods, exposure and outcome assessment, results, and discussion. A commendable effort by the team.

---

## Round 0.2 · accepted · Accept

After revisions, three reviewers agreed to publish the manuscript. I also reviewed the manuscript and found no obvious risks to publication. Therefore, I also approved the publication of this manuscript.

Reviewer 2 ·

Basic reporting

Self contained with relevant results to hypotheses

Experimental design

methods described with sufficient detail and information to replicate

Validity of the findings

Conclusions are well stated, linked to original research question & limited to supporting results.